# Evaluation of NO_x_ and PN Emission in Relation to Actuator Control

**DOI:** 10.3390/s24144430

**Published:** 2024-07-09

**Authors:** Norbert Biró, Dániel Szőllősi, Péter Kiss

**Affiliations:** 1IBIDEN Hungary Kft. Technical Center, Exhaust System Evaluation, 2336 Dunavarsany, Hungary; szollosi.daniel.1@phd.uni-mate.hu; 2Department of Vehicle Technology, Institute of Technology, Hungarian University of Agriculture and Life Sciences, 2100 Godollo, Hungary; kiss.peter@uni-mate.hu

**Keywords:** 10 and 23 nm particle emission, NO_x_ emission, EGR, rail pressure, diesel vehicles, emissions evaluation, Euro 7

## Abstract

This study aimed to investigate the interrelationships between key harmful emission components, nitrogen oxides (NO_x_), and particulate numbers (PNs) in diesel engine exhaust and the control actuators of diesel engines. This research involved conducting a series of experiments under fixed parameters within an engine brake laboratory environment to elucidate these correlations. The objectives of this study were to conduct a comprehensive review of the relevant emissions technology literature and a comparative assessment of particle measurement methods based on dilution ratios and develop innovative aerosol preparation principles tailored to condensation particle measurement. Additionally, this research involved designing and implementing an aerosol preparation unit based on the newly developed principles, along with the creation of test cell control programs using the AVL PUMA Open TST editor interface and Visual Basic. Furthermore, this study was concerned with conducting evaluations of fixed-parameter engine dynamometer tests to explore the functional relationships between the emission of 10/23 nm particles, NO_x_ emissions, common rail pressure variations, and exhaust gas recirculation levels. This study aimed to enhance the understanding of diesel engine emissions dynamics and contribute valuable insights for developing more efficient and environmentally friendly engine control strategies.

## 1. Introduction

Emissions technology represents a rapidly evolving sector within the automotive industry of the 2000s, driven by increasingly stringent emissions regulations imposed at both European and global levels. This necessitates the development of advanced exhaust gas aftertreatment systems to comply with these regulations effectively. The Volkswagen diesel scandal in 2015 brought heightened public attention to the need for the stringent regulation of tailpipe emissions, emphasizing the importance of considering all potential scenarios at the design stage to minimize environmental impact [1]. In response to the regulatory landscape, there is a clear push towards the development of newer and more sophisticated exhaust gas aftertreatment systems to address emissions challenges. This includes technologies such as selective catalytic reduction (SCR), diesel particulate filters (DPF), and lean NO_x_ traps (LNT), among others, which play crucial roles in reducing harmful pollutants from vehicle exhaust fumes. Recent studies [2] have highlighted the continued significance of both compression ignition and spark ignition internal combustion engine-based hybrid vehicles in the automotive market. This underscores the ongoing relevance of research aimed at evaluating emissions from internal combustion engines, especially in the context of evolving regulatory frameworks and consumer expectations. The specific focus of this study was to investigate the relationship between particle and NO_x_ emissions [3] from a Euro 6-d heavy-duty diesel engine in relation to key parameters such as exhaust gas recirculation (EGR) and common rail pressure. Measurements were conducted on an engine dynamometer under controlled conditions to ensure consistency and accuracy. This included maintaining fixed temperatures for external fluids (coolant, engine oil, fuel) and intake air throughout the testing process. Modern diesel engine control units exhibit sophisticated adaptive strategies [4], adjusting operational modes and strategies in response to varying engine loads and ambient conditions. While this adaptability is advantageous in real-world driving scenarios, it presents challenges in laboratory settings where achieving fixed-parameter, repeatable measurements is essential for rigorous scientific analysis [5]. To address the need for engine actuator control during testing, a custom Visual Basic was developed to manage engine actuators using Accurate Technologies (ATI) Vision software v6.0 in real time. This approach allowed for the precise manipulation of specific actuators while keeping others at fixed values, ensuring consistency and reliability in experimental results. Looking ahead to the implementation of Euro 7 regulations, this study places particular emphasis on measuring particle number emissions in the 23 nm and 10 nm ranges [6]. This comparative analysis provides insights into the potential impact of Euro 7 standards on emissions severity and compliance, thereby informing future advancements in emissions control technologies and regulatory strategies within the automotive industry.

## 2. Materials and Methods

### 2.1. Test Cell Infrastructure

Gaseous and liquid media (coolant, oil, intake air, and charge air) significantly influence engine behavior [7]. Conditioning systems are capable of maintaining these media at predetermined temperatures under all operating conditions. The execution of highly accurate and repeatable tests necessitates fixed environmental conditions. Exhaust gas emission measurement tests were conducted with a time delay, following the evaluation of each test; thus, the conditioning of the media ensured that the testing conditions remained comparable. For these reasons, conditioning systems were indispensable for the experiments intended to be carried out. The experiments were conducted in the research and development center of Ibiden Hungary Ltd. located in Hungary, where the mentioned conditioning devices and complete engine dynamometer test cells can be found. A summary of the testing apparatus is listed in Table 1.

The engine dynamometer test cell is illustrated in Figure 1.

For the evaluation of test cycles, appropriate sampling times for data acquisition are also crucial. The data acquisition apparatus included the operational parameters of the engine (such as rotation speed and engine torque), environmental conditions (such as test cell air temperature, intake air pressure, humidity, temperature, engine fluids’ temperatures, pressure, etc.), and the specific amounts of gaseous and solid pollutants. Additionally, data from the engine’s on-board diagnostics (OBDs) sensors were recorded, while also having the ability to modify the actuators that were fundamentally regulated by the engine control unit.

### 2.2. Test Engine

For the experiments, a Volvo-made, common rail, direct injection diesel engine designed for heavy-duty vehicles was utilized, which is shown in Figure 2. The engine used was parametrized, allowing for modifications in its key characteristics. This enabled the application of specific settings on the engine, regardless of the engine type, thereby ensuring that the engine subsystems operated with fixed parameters while the parameter under examination was varied. As a result, reproducible measurements were conducted, independent of the engine type, which facilitated the generalizability and repeatability of the research findings. The engine data can be found in Table 2.

### 2.3. Experiment Planning

#### 2.3.1. WHSC Cycle

The World Harmonized Steady State Cycle (WHSC) is a constant-state engine dynamometer test cycle defined by Regulation No. 4 of the Global Technical Regulation (GTR) [8]. The WHSC, which includes steady-state engine conditions, was developed to cover the characteristic driving habits of the EU, USA, Japan, and Australia. The WHSC test cycle is a key part of the complex certification framework aimed at harmonizing heavy-duty vehicle emissions on a global scale. Therefore, the WHSC cycle was chosen for this research because it allowed the results to be easily compared with data from manufacturers and other researchers.

The so-called normalized values defined the percentages of the maximum engine speed and torque. In contrast, denormalized values were derived specifically for the given engine. The denormalized engine speed was determined for the test engine with the help of Equation (1) [9].

Denormalization calculation:

*n_ref* = *n_norm*·(0.45·*n_lo_* + 0.45·*n_pref_* + 0.1·*n_hi_* − *n_idle_*)·2.0327 + *n_idle_*
(1)


*n_lo_* is the lowest rotational speed at which the engine’s output reaches 55% of its maximum power.*n_pref_* is the engine speed at which the integral of the mapped torque between *n_idle_* and *n_95h_* constitutes 51% of the total integral. This speed indicates the most favorable torque characteristics within the operational range.*n_hi_* is the highest speed at which the engine’s performance achieves 70% of its maximum power, marking an upper limit of efficient power output.*n_idle_* is the idle speed of the engine*n_95h_* Is the highest speed at which the engine’s output reaches 95% of its maximum power, defining the upper performance limit under high load conditions.

#### 2.3.2. AVL PUMA Programming Environment

AVL PUMA software v1.5.1 is associated with test cells manufactured by AVL GmbH. The entire system adheres to the “single software principle”, meaning that control of the previously described conditioning systems, the engine dynamometer, the engine itself, the emission measuring equipment, the fuel system, data acquisition, and PLCs (Programmable Logic Controllers) found at the facility operation level was accessible from a control computer located in front of the test cells. This integration facilitated the comprehensive management and synchronization of test cell operations, enhancing efficiency and precision in the testing processes.

The WHSC mode 9 and WHSC custom test programs provide strictly fixed parameters during experiments. WHSC mode 9 serves as an engine pre-conditioning before WHSC cycles. Each experiment is executed by the control program in exactly the same sequence and steps (as shown in Figure 3), ensuring the reliability and repeatability of the results. Achieving this level of precision with manual control would be impossible [10]. Through the use of consistent and standardized procedures during the test cycles, the experiments were comparable and reproducible, which is crucial for obtaining reliable results and conclusions in engine testing.

### 2.4. Experiment Schedule

The experiments were conducted at the Technical Center located at the Ibiden Hungary facility, utilizing the equipment described in previous sections. Engine control programs were developed in Visual Basic to forcibly modify the different actuator values of the engine. During the operation of the engine, all parameters were fixed, meaning that only the actuator values modified for the specific experiments deviated from their baseline state. This was guaranteed by the uniquely created dynamometer and engine control automated programs, ensuring that each experiment was perfectly consistent in terms of the desired parameters. There were 12 different variants of the experiments (shown in Table 3), each repeated 3 times, resulting in a total of 36 test series. Moreover, since each test series was performed, comprising one WHSC mode 9 preconditioning and one WHSC main program, a total of 72 separate engine dynamometer measurements were conducted. A single test series took exactly 4.08 h, meaning that the entire measurement process spanned 150 h, which, calculated over 8-h workdays, extended over more than 3.5 weeks.

During the testing, the rail pressure and EGR setpoints were modified: 0% would mean the original actual setpoint, which was determined by the ECU’s predefined map. The setpoints were adjusted in the range of −30% to +30%. During every test, only the observed actuator was adjusted, whereas others were fixed.

### 2.5. Emission Measurement Devices

#### 2.5.1. Particle Counter (APC 489)

Condensation Particle Counting (CPC) is the most widely used [11] method for determining the particle number concentration of an aerosol (sample gas). This technique involves the treatment of the aerosol’s solid particles with liquid butanol, causing them to enlarge. These enlarged particles pass through an extremely sensitive sensing chamber, equipped with laser light and sophisticated photodetection optics, designed for precise particle analysis and quantification. This method enables the accurate detection and counting of particles, even at very low concentrations, making it indispensable for assessing air quality, environmental monitoring, and compliance with emissions standards.

#### 2.5.2. Gas Analyzer (AMA i60)

The majority of exhaust gas components regulated by the Euro vehicle emission directives consist predominantly of gaseous pollutants. The analyzers used during the AMA i60 experiments had a wide measurement range and were capable of accurately measuring gaseous pollutants such as THC, NO/NO_2_/NO_X_, CO, CO_2_, O_2_, N_2_O, CH_4_, SO_2_, and NH_3_. Since the measurement principles of various gaseous emission components differ from one another, aerosols must be analyzed using different analyzers, which are separated using a common measurement tube. From a research perspective, NO_x_ as an emission component is of particular importance [12], with its associated analyzer based on the principle of the chemiluminescence measurement. This method is critical for the accurate detection and quantification of NO_x_ emissions, offering valuable insights into the environmental impact of vehicle exhaust.

#### 2.5.3. Measurement Uncertainty

The AVL APC489 and AMA i60, both products of AVL List GmbH, Graz, Austria, are subjected to an annual calibration process executed by the original equipment manufacturer. This calibration process is uniformly applied to all units. The calibration of particle number is conducted using a condensation particle counter as the benchmark, while the gas analyzer is calibrated utilizing pure synthetic gas as the standard reference. The verification of measurement uncertainty is conducted in accordance with the standards set by “ISO/IEC 17025” [9]. The overall certainty of the measurements is at least 95%.

### 2.6. Data Analysis with Linear and Polynomial Regression

The analyzed dataset was cleaned of outliers using polynomial and linear regression. During the regression process, the removal of outliers aided in enhancing the stability and reliability of the model. This method allowed for a detailed examination of the interactions between variables and their impact on the dependent variable, minimizing the potential distorting effect of outliers on the analysis. By employing these regression techniques, it was possible to better understand the underlying relationships within the data, providing insights that were critical for accurate predictions and interpretations in the context of this research.

### 2.7. Calculation Method for PN and NO_x_

To determine the particle count, it was essential that the exhaust gas volumetric flow rate be determined. However, since the measured results were recorded proportionally by mass, conversion was necessary. The calculations were based on the ideal gas law, also known as the Clapeyron equation [13], which combines Boyle’s–Mariotte, Gay-Lussac, and Avogadro’s empirical laws (2).
(2)p·V=n·R·TThe molar count was substituted (3) into the ratio of the gas’s total and molar masses. In mathematical terms, this is as follows:(3)$$p\cdot V=\frac{m}{M}\cdot R\cdot T$$Subsequently, the equation was arranged (4) with respect to the gas pressure.
(4)$$p=\frac{m}{V}\cdot \frac{R}{M}T$$Since the universal gas constant and the ratio of molar mass provided the specific gas constant for the gas, the equation could be simplified (5) as follows:(5)$$p=\frac{m}{V}\cdot R_{S}\cdot T$$To calculate the particle count of a solid, it is necessary to apply the volumetric flow rate; thus, the equation was arranged with respect to volume (6).
(6)$$V=\frac{m\cdot R_{S}\cdot T}{p}\;\left[{cm}^{3}\right]$$To perform accurate calculations, it was first necessary to determine (7) and (8) the total mass of air ingested by the engine and the consumed fuel, based on mass flow rate.
(7)$${EMF}_{hr}=GAH+{FB}_{val}\;\left[\frac{kg}{h}\right]$$
(8)$${EMF}_{sec}=\frac{{EMF}_{hr}}{3600}\;\left[\frac{kg}{s}\right]$$For further calculations, the specific gas constant of the air needed to be determined (9).
(9)$$R_{air}=\frac{R}{M}=\frac{8.314}{0.02897}=287\;\left[\frac{J}{kgK}\right]$$Subsequently, the equation derived from the gas law (4) was utilized to determine (10) the volumetric flow rate.
(10)$$EVF=\frac{{EMF}_{sec}\cdot R_{air}\;\cdot T}{p}\;\left[\frac{{cm}^{3}}{s}\right]$$The variables were substituted and the equation was rearranged with respect to volumetric flow rate (11).
(11)$$EVF=\frac{{EMF}_{sec}\cdot 287\;\cdot 273}{\left(101.3\cdot 10^{3}\right)\cdot 10^{6}}\;\left[\frac{{cm}^{3}}{s}\right]$$Finally, exhaust volume flow was determined (12) for the whole length of the cycle.
(12)$${EVF}_{sum}=\sum_{0}^{1894}EVF\left[{cm}^{3}\right]$$The 10 and 23 nm particle counts were calculated based on the Equations (13) and (14) below.
(13)$${SPN}_{10}=\frac{{EVF}_{sum}\;\cdot {PN}_{10}}{W_{WHSC}}\;\left[\frac{\#}{kWh}\right]$$
EVFsum: exhaust volume flow under the WHSC cycle [cm^3^];PN10: average particle count from 10 nm, under the WHSC cycle [#/cm^3^];$W_{WHSC}$: work performed under the WHSC cycle [kWh];SPN10: solid particle count from 10 nm [#/kWh].
(14)$${SPN}_{23}=\frac{{EVF}_{sum}\;\cdot {PN}_{23}}{W_{WHSC}}\;\left[\frac{\#}{kWh}\right]$$EVFsum: exhaust volume flow under the WHSC cycle [cm^3^];PN23: average particle count from 23 nm, under the WHSC cycle [#/cm^3^];$W_{WHSC}$: work performed under the WHSC cycle [kWh];SPN23: solid particle count from 23 nm [#/kWh].
The Equation (5) was used to determine the density of ideal gases.
(15)$$\rho =\frac{p}{R_{S}\cdot T}$$
ρ: ideal gas density [g/dm^3^];p: ideal gas pressure [kPa];R_s_: ideal gas specific constant [J/gK];T: ideal gas temperature [K].
The gaseous emissions were calculated in $\frac{g}{{dm}^{3}}$. The below equations were calculated with atmospheric pressure and at 273 K.

Nitrogen ideal gas density calculations (16)–(18):(16)$$\rho_{N_{2}\;}=\frac{p}{R_{N_{2}}\cdot T}\;$$
(17)$$\rho_{N_{2}\;}=\frac{101.325}{0.2968\cdot 273}=1.2342\;\left[\frac{g}{{dm}^{3}}\right]$$
(18)$$\rho_{N}=\frac{\rho_{N_{2}\;}}{2}=0.6171\;\left[\frac{g}{{dm}^{3}}\right]$$Oxygen ideal gas density calculations (19)–(21):(19)$$\rho_{O_{2}\;}=\frac{p}{R_{O_{2}}\cdot T}\;$$
(20)$$\rho_{O_{2}\;}=\frac{101.325}{0.2598\cdot 273}=1.428\;\left[\frac{g}{{dm}^{3}}\right]$$
(21)$$\rho_{O}=\frac{\rho_{O_{2}\;}}{2}=0.714\;\left[\frac{g}{{dm}^{3}}\right]$$The term NO_x_ serves as a collective name encompassing nitrogen oxides. In most cases, it specifically refers to nitrogen dioxide (NO_2_). Therefore, the calculations (22) and (23) were limited to provide the ideal gas density of NO_2_.
(22)$$\rho_{{NO}_{x}\;}={(\rho }_{N}+\rho_{O_{2}\;})+(\rho_{N}+{\rho_{O}}_{\;})$$
(23)$$\rho_{{NO}_{x}\;}=2.0451+1.3311=3.3762\;\left[\frac{g}{{dm}^{3}}\right]$$In order to determine the weight ratio, the air density was necessary, which could be calculated with the ideal gas laws (24) and (25).
(24)$${\rho_{\;}}_{air}=\frac{p}{R_{air}\cdot T}$$
(25)$${\rho_{\;}}_{air}=\frac{101.325}{0.287\cdot 273}=1.2932\left[\frac{g}{{dm}^{3}}\right]$$The previously determined (22) NO_x_ density enabled the calculation of the mass flow ratio of NO_x_ in the exhaust gas (27). The following equations were applied for dry conditions.
(26)$${NO}_{x_{m}}=\frac{{EMF}_{hr}}{\rho_{air}}\cdot \frac{\rho_{{NO}_{x}\;}\cdot 10^{6}}{3600\cdot 10^{3}\cdot {NO}_{x_{raw}}}\left[\frac{g}{s}\right]$$
(27)$${NO}_{x_{m}}=\frac{{EMF}_{hr}}{1.2932}\cdot \frac{3.3762\cdot 10^{6}}{3600\cdot 10^{3}\cdot {NO}_{x_{raw}}}\left[\frac{g}{s}\right]$$Next, the summation of NO_x_ weight at every second of the cycle was determined using the equation below (28).
(28)$${NO}_{x_{m\;sum}}=\sum_{0}^{1894}{NO}_{x_{m}}\left[g\right]$$Finally, NO_x_ weight was divided by the work performed under the WHSC cycle, as shown below (29).
(29)$${NO}_{x_{P}}=\frac{{NO}_{x_{m\;sum}}}{WW_{WHSC}}\left[\frac{g}{kWh}\right]$$

## 3. Results

### 3.1. Analysis of Particle Counting Lab Equipment

The topic of particle number measurement was of paramount importance during the experiments. For this purpose, before starting the series of experiments, two particle counter setups were tested. The focus of the conducted measurements was the simultaneous examination of the measurement range between 10 and 23 nm. This became particularly important since the current Euro VI standard mandates counting particles in exhaust gas starting from a 23-nm particle diameter [14]. However, the upcoming Euro VII standard will require the counting of particles with diameters from 10 nm. This development was an important step because it ensured that the research findings not only comply with current standards but also remain relevant and useful following the introduction of Euro VII. Measuring particles as small as 10 nm helps guarantee that the research results and information related to particle concentration will continue to be valuable and applicable in the future with the emergence of new standards and requirements. Due to the inherently different measurement principles of the Advanced Particle Counter (APC) and the Electrical Exhaust Particle Sizer (EEPS), initial tests were necessary to verify that the two devices could measure together accurately. The validating measurements were conducted with the WHSC cycle, the results of which are shown in Table 4 and presented in Figure 4, where the dilution factor for the final particle number result was calculated for each WHSC measurement.

The green area on the chart indicates where the two devices were able to measure effectively in conjunction. The overlap, where the counting efficiency of both devices was acceptable, was quite small and not deemed suitable for extensive experimental work; therefore, the first particle counting setup was not optimal. In contrast, the second setup, where the two devices operated almost entirely with a matching measurement methodology, allowed for efficient measurement across the entire blue field. Additionally, as subsequent experiments were based on the measurement of raw exhaust gas, a higher aerosol dilution was necessary to protect the particle counting equipment. Furthermore, the dilution capacity of the EEPS below 100 was entirely unusable for engine out (EO) measurements, further reinforcing the use of the second setup. The use of the EEPS would have provided a more comprehensive picture of particle formation by determining the distribution of particles across different size classes. We ultimately decided against its further application in the experiments ahead. The experiment and its results were also published in 2023 and discussed more extensively [15].

### 3.2. Development of the Particle Counter

Before the start of the planned series of engine dynamometer emission measurement experiments, certain challenges had to be addressed associated with condensation particle measurement. Specifically, in lengthy experimental series that occur in measurement environments with significant particle number concentrations and high relative humidity—typically situations where the emission measuring systems operate without aftertreatment devices [16]—clogging of the aerosol (sample gas) flow paths and condensate formation can occur. These phenomena can lead to the malfunctioning of measuring equipment, hindering the execution of the planned nearly 150-h experimental series. To avoid this, a new aerosol preparation unit had to be designed aimed at preventing the clogging of sample lines and the formation of condensate, thereby ensuring the successful completion of the experimental series. This approach underscores the importance of addressing operational challenges to maintain the integrity and accuracy of long-duration emission measurement studies. As with many studies focusing [17,18,19,20] on sub-23 nm tests, reliable and repeatable measurements were important.

The developed aerosol preparation unit incorporated new components and subunits, including a condensate drainage unit, a dedicated container for collecting condensed liquids, a high-efficiency particulate air (HEPA) filter, an adjustable power supply, and a membrane pump, as shown in Figure 5. The flow of sample gas through the developed unit is illustrated in Figure 6. This enhanced design aimed to address the challenges of aerosol preparation in environments with high particle concentrations and humidity, ensuring reliable measurement performance and preventing equipment malfunction through innovative engineering solutions. The particle counter development and its validation results were also published in 2023 and discussed more extensively [21].

### 3.3. Evaluation of Experiments

The recorded data, due to the large number of repetitions, had to be cleansed with cluster analyses. Adjustments to the exhaust gas recirculation and common rail pressure parameters were made symmetrically in relation to the baseline conditions, ranging from −30% to +30%, in 10% increments. In the following sections of this chapter, the experimental data, which had been already filtered by cluster analyses, will be shown and discussed further. Given that a single WHSC cycle captures approximately 160,000 data points, the 12 experiments required the processing of nearly 2 million data points in total. To manage the data reliably, an automated data processing method had to be employed, which was developed in a Microsoft Excel format. To maintain the readability of this article and reduce its length, only one of the 12 experiments will be presented.

Figure 7 illustrates that reducing the extent of exhaust gas recirculation (EGR) resulted in a significant increase in the values of nitrogen oxide (NO) and nitrogen dioxide (NO_x_), while the level of carbon dioxide (CO_2_) decreased. This was attributed to the fact that the EGR method used for NO_x_ reduction adversely impacted the engine’s thermal efficiency. Therefore, when the EGR rate was reduced by 30% in this experiment, CO_2_ emissions and, consequently, fuel consumption decreased, while the work output increased, as identified by Needham and Park [22,23]; Mei [24] also derived similar conclusions.

In the final phase of the evaluation, the pressure in the common rail system and the extent of exhaust gas recirculation (EGR) throughout the cycle were examined. These parameters were controlled through the engine management system with the help of a specifically developed program code. In this experiment, the EGR opening value was reduced by 30%, which can be clearly followed in Figure 8.

### 3.4. Functional Relationship between the Emission of 10/23 nm Particles and Common Rail Pressure

The first goal was to investigate the number of solid particles in the exhaust gas of a diesel engine as a function of changes in the common rail pressure. During this study, two particle sizes were distinguished: 10 nm and larger and 23 nm and larger. For modeling the relationship, polynomial regression was chosen because a quadratic relationship could be observed between the data, a point also referenced in Siebers’s research [25]. The polynomial regression model allowed for the description of non-linear patterns as well. Furthermore, it met the predefined objective of at least a 97% fit to the experimental data points, based on preliminary calculations. The regressions are presented in Figure 9.

### 3.5. Functional Relationship between the Emission of 10/23 nm Particles and the Opening Ratio of Exhaust Gas Recirculation

The second goal of this research was to investigate the number of solid particles found in the exhaust gas of a diesel engine as a function of the percentage adjustment of exhaust gas recirculation (EGR). During this study, two particle sizes were distinguished: those 10 nm and larger and those 23 nm and larger. For modeling the relationship, polynomial regression was chosen. In this case, linear regression would have worked as a good approximation based on a preliminary study; however, it could not meet the set error margin of 3%. The functions and measurement points are illustrated in Figure 10. The results are comparable to O’Connor [25] and Kuzuyama’s [26] studies. Fuyuto’s conclusions also can be related [27].

### 3.6. Functional Relationship between NO_x_ Emissions and the Opening Ratio of Exhaust Gas Recirculation

The third goal of the research was to examine the relationship between changes in the opening value of EGR and the nitrogen oxide emissions (NO_x_) of a diesel engine. The EGR opening was adjusted with the settings of the tested engine by a fixed amount: from −30% to +30%, in 10% intervals. Concurrently, NO_x_ emission levels were calculated in g/kWh. For the mathematical modeling of the relationship, a linear regression was used for the analysis. A regression model was fitted to the data, which described the linear relationship between the independent variable (*x*) and the dependent variable (*y*).

The function is visually depicted in Figure 11. The results were in line with other studies in similar fields [28,29].

### 3.7. Functional Relationship between NO_x_ Emissions and Common Rail Pressure

The fourth objective was to examine the relationship between changes in common rail pressure and the nitrogen oxide emissions (NO_x_) of a diesel engine. The common rail pressure settings of the tested engine were adjusted by a fixed amount: from −30% to +30%, in 10% intervals. Concurrently, the NO_x_ emission levels were calculated in g/kWh. For the mathematical modeling of the relationship, a linear regression analysis had to be carried out. The regression is presented in Figure 12. Shrivastav’s [30] CFD analyses showed close similarity to the current experiment’s empirical results.

## 4. Discussion

Special attention was given to the complex problem of fixed parameters from the external and engine side as well throughout the research. Several challenges had to be addressed at the start, including understanding the selection and dilution-dependent operation of particle counters, designing an aerosol preparation unit that ensured reliable particle measurements during long test series, and developing program codes for engine actuators to allow precise control. Innovative procedures developed in response to these challenges enabled us to conduct over 300 h of experiments with fixed parameters from the engine side, the results of which were described by regression models with high determination coefficients (R^2^ > 0.97). These results accurately approximate the two main emission components, NO_x_ and PN, of newly developed heavy-duty vehicles equipped with compression ignition, common rail fuel injection systems, and electronically controlled exhaust gas recirculation systems meeting the Euro VI standard, as they relate to changes in common rail pressure and the extent of exhaust gas recirculation. With regressions determined for particulate number concentrations across the measurement ranges of 10 and 23 nm, these can be utilized in upcoming research related to the soon-to-be-finalized EURO 7 standard. Thus, the model developed served not only the current research objectives but also provides a solid foundation for the creation of more complex models. These advanced models can offer deeper and more comprehensive insights into the emission dynamics and behavior of diesel engines, enabling the optimization of engine performance while complying with environmental regulations.

## Figures and Tables

**Figure 1 sensors-24-04430-f001:**
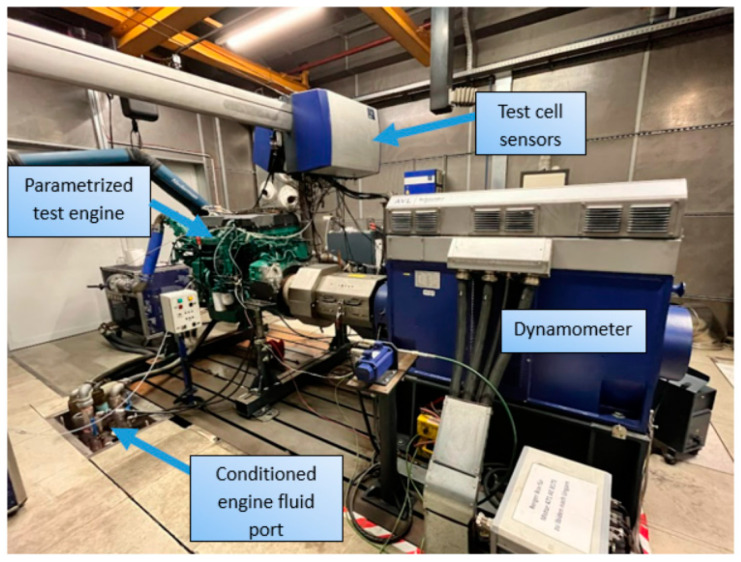
Engine dynamometer test cell at Ibiden Hungary Ltd.’s facility. [Source: Norbert Biro].

**Figure 2 sensors-24-04430-f002:**
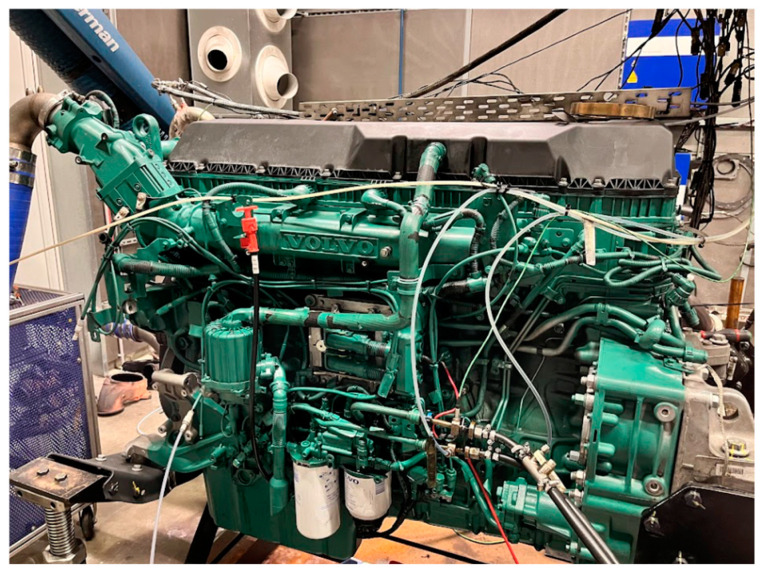
Volvo D13-TC installed on an AVL HD 500 kW engine dynamometer at the Ibiden Hungary Technical Center. [Source: Norbert Biro].

**Figure 3 sensors-24-04430-f003:**
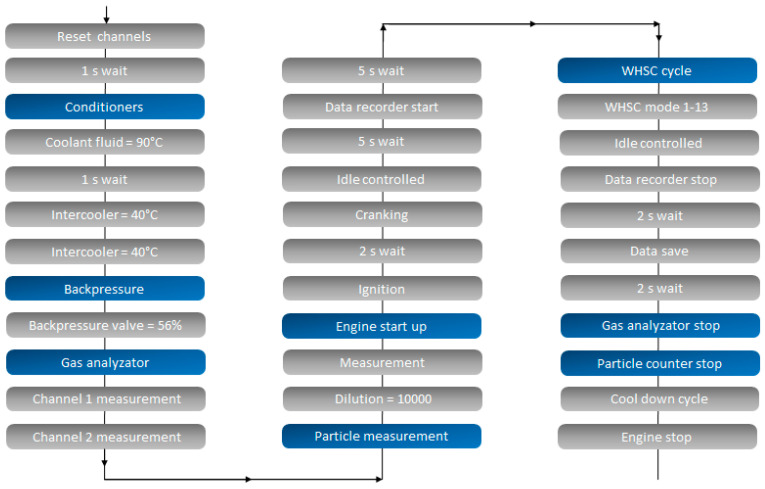
Steps of the WHSC program on the AVL Puma programming interface. [Source: Norbert Biro].

**Figure 4 sensors-24-04430-f004:**
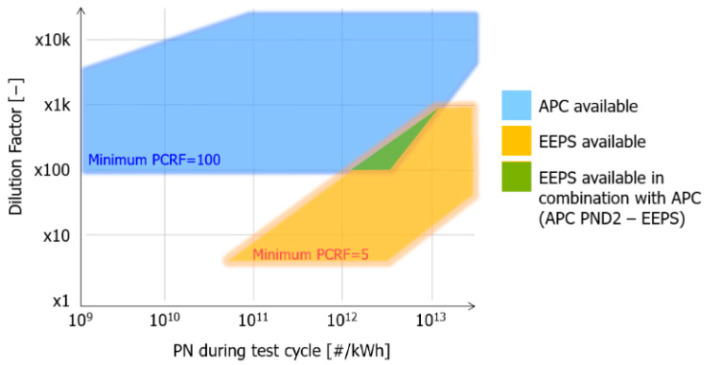
Results of particle number measurements with the APC and EEPS as a function of the dilution factor. APC = AVL Particle Counter, CPC = Condensation Particle Counter, and EEPS = Exhaust Emission Particle Sizer. [Source: Norbert Biro] [15].

**Figure 5 sensors-24-04430-f005:**
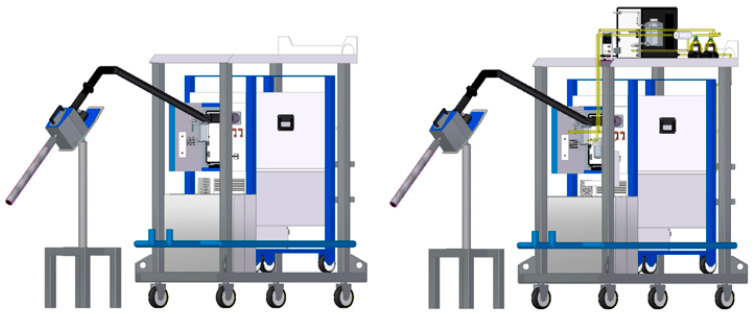
CAD models of the original (**left**) and developed (**right**) aerosol preparation units for a condensation particle counter. [Source: Norbert Biro] [21].

**Figure 6 sensors-24-04430-f006:**
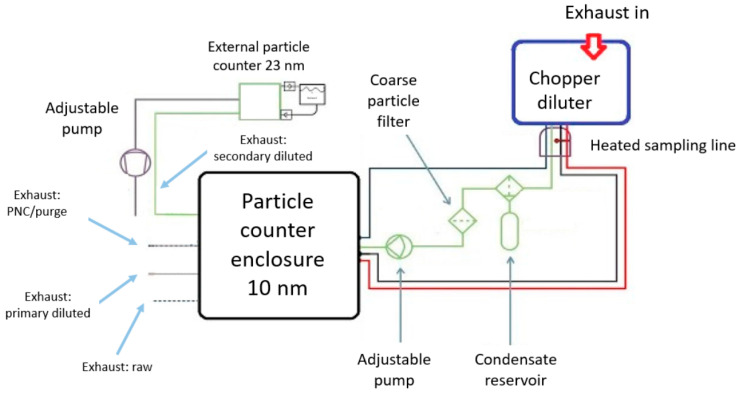
Flow diagram of the APC 489, with the N-configuration. [Source: Norbert Biro] [21].

**Figure 7 sensors-24-04430-f007:**
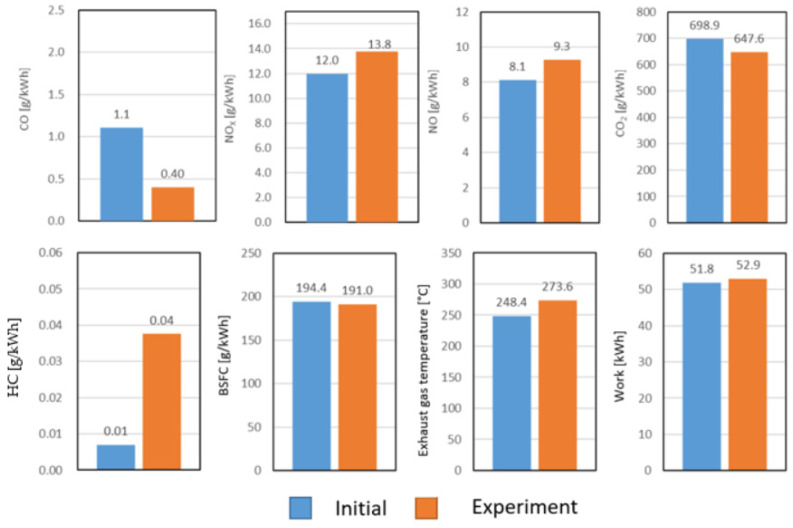
Calculated carbon monoxide, nitrogen oxide, nitrogen monoxide, carbon dioxide, hydrocarbons, specific fuel consumption, exhaust gas temperature, and work performed during the experimental WHSC cycle, compared with the baseline. [Source: Norbert Biro].

**Figure 8 sensors-24-04430-f008:**
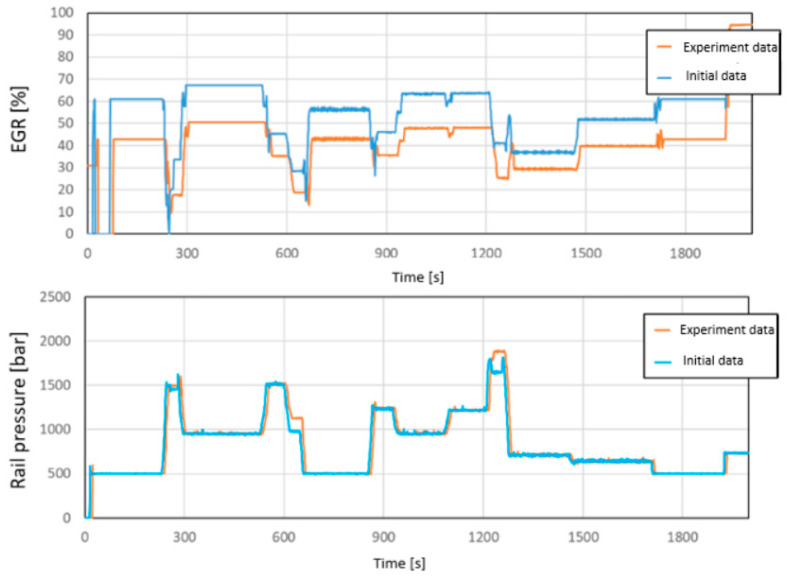
Transient exhaust gas recirculation ratio (EGR) and common rail pressure during the experimental WHSC cycle, compared with the baseline. [Source: Norbert Biro].

**Figure 9 sensors-24-04430-f009:**
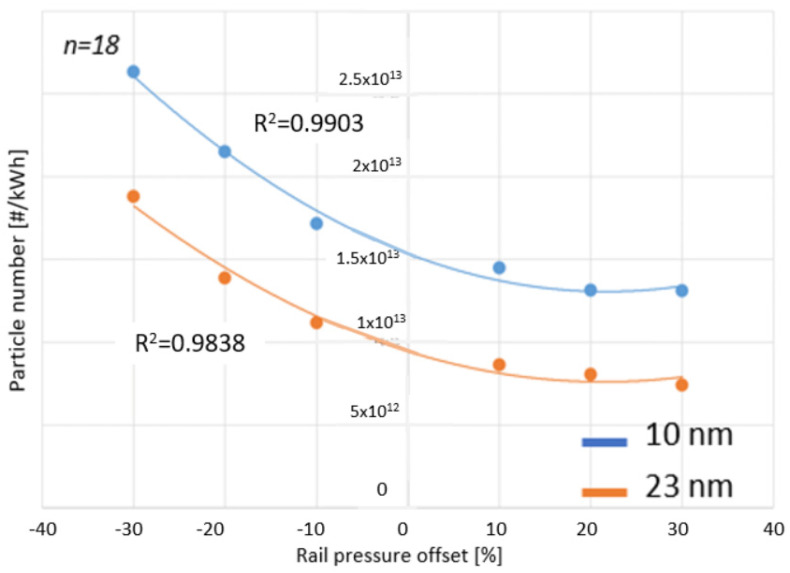
Particle numbers of 10 and 23 nm as a function of common rail pressure changes during the WHSC cycle. [Source: Norbert Biro].

**Figure 10 sensors-24-04430-f010:**
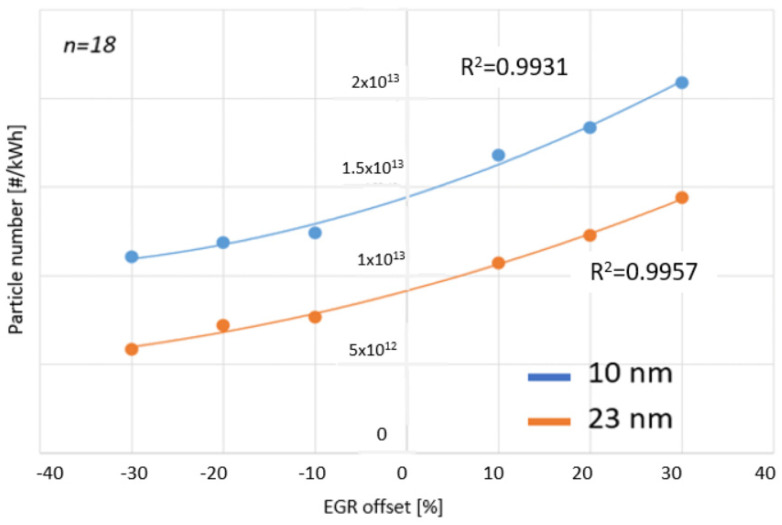
Particle numbers of 10 and 23 nm as a function of changes in exhaust gas recirculation during the WHSC cycle. [Source: Norbert Biro].

**Figure 11 sensors-24-04430-f011:**
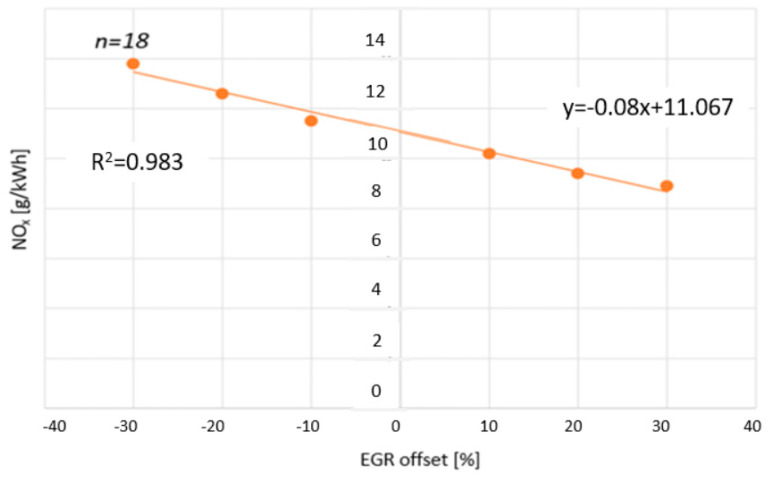
NO_x_ concentration as a function of changes in exhaust gas recirculation during the WHSC cycle. [Source: Norbert Biro].

**Figure 12 sensors-24-04430-f012:**
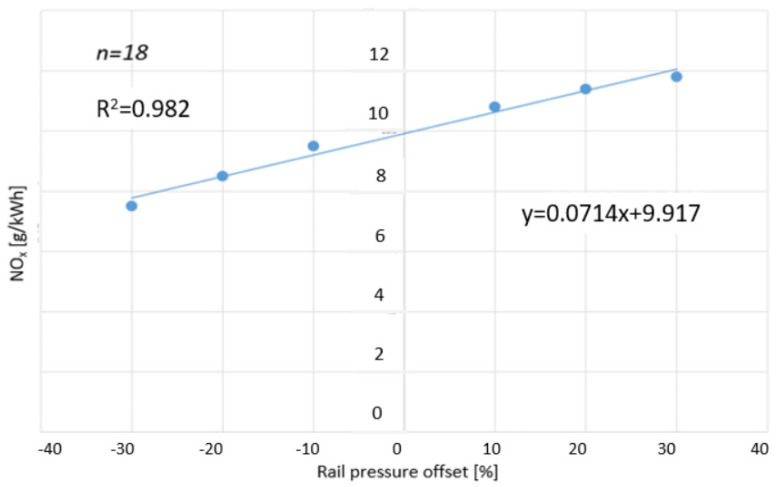
NO_x_ as a function of changes in common rail pressure during the WHSC cycle. [Source: Norbert Biro].

**Table 1 sensors-24-04430-t001:** Summary of the testing apparatus.

Dynamometer Model	AVL HD 500 kW (INDY S50-4/3001-1BS-1)
Test enviroment control system	AVL Puma Open 1.5.1
Coolant media conditioner	AVL ConsysCool
Lubricant conditioner	AVL ConsysLube
Intake air conditioner	AVL ConsysAir
Charge air conditioner	AVL ConsysBoost
Particle counter	AVL APC 489
Gas analyzer	AVL AMA i60

**Table 2 sensors-24-04430-t002:** Test engine specifications.

Engine Specification	
Engine model	D13TC TURBO-TC
Injection type	Direct injection diesel
Bore × stroke	131 [mm] × 158 [mm]
Displacement	12,800 [cm^3^]
Firing order	1-5-3-6-2-4
Performance	372 [kW] @1300-1600 [1/min]
Torque	2840 Nm @900-1300 [1/min]
Compression ratio	18:01

**Table 3 sensors-24-04430-t003:** Test types.

#	Actuator	Offset	Denomination
1	EGR	-30%	EGR-30%
2	RailP	-30%	RailP-30%
3	EGR	-20%	EGR-20%
4	RailP	-20%	RailP-20%
5	EGR	-10%	EGR-10%
6	RailP	-10%	RailP-10%
7	EGR	10%	EGR+10%
8	RailP	10%	RailP+10%
9	EGR	20%	EGR+20%
10	RailP	20%	RailP+20%
11	EGR	30%	EGR+30
12	RailP	30%	RailP+30%

**Table 4 sensors-24-04430-t004:** Results of experiments conducted with APC and EEPS particle counters with varying dilution factors. [Source: Norbert Biro].

Test nr.	APC DILUTION Factor [-]	Calculated Particle Count [#/kWh]	EEPS Dilution Factor [-]	Calculated Particle Count [#/kWh]
1	100	5 × 10^12^	5	5 × 10^12^
2	5000	5 × 10^13^	50	5 × 10^13^
3	15,000	5 × 10^13^	1000	5 × 10^13^
4	100	1 × 10^9^	5	8 × 10^10^
5	5000	1 × 10^9^	50	5 × 10^11^
6	15,000	1 × 10^11^	1000	1 × 10^13^

## Data Availability

The data presented in this study are available upon request from the corresponding author. The data are not publicly available as research data could aid competitors of the company where the experiments were carried out.

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
