# Peer review of "Evaluation of NO_x_ and PN Emission in Relation to Actuator Control"

_sensors, 2024, doi:10.3390/s24144430_

Round 1
Reviewer 1 Report
Comments and Suggestions for Authors
Dear Authors,
In order to improve the quality of the article, I suggest including specific issues:
1) Materials and methods: It would be beneficial to complete, in addition to a photograph of the test stand (in Figure 1), a detailed diagram of the test stand with a description of the apparatus used in the tests. The description of “engine” in Figure 1 should also be improved.
2) Materials and methods: Line 84 – The first person “I had...” should be avoided.
3) Materials and methods: In lines 114-115, the authors refer to equation (1) in terms of speed and torque. This equation applies only to rotational speed. In the equation, the value “2,0327” should be corrected as “2.0327”. In the description to equation (1), the nidle (lower case) should be corrected in line 126.
4) Materials and methods: Figure 3: For what purpose is the Intercooler=40°C phase repeated?
5) Materials and methods: Line 141: For what purpose do the authors refer to WHSC mode 9?
6) Materials and methods: Line 161: Was the study implemented at 3500 weeks or 3.5 weeks?
7) Materials and methods: It would be beneficial to improve the quality of Table 2.
8) Materials and methods: Complete the designations of the quantities used in the equations.
9) Materials and methods: In all equations, the decimal separator should be corrected to a dot.
10) Materials and methods: Why do the authors take NO2 as NOx in the calculation methodology (equation (23))? They omit NO, which is also measured as a component of NOx, particularly since it is the dominant contributor to NOx (Figure 7).
11) Materials and methods: Equations (26) and (27) lack NOx concentrations measured in the exhaust gas. In addition, the NOx values are multiplied by a humidity correction factor, which the authors do not even mention. Why?
12) Materials and methods: The authors do not provide any information on the uncertainty of the measurements. This point should be completed in the study.
13) Results: Figure 6 – The descriptions should be improved. They are unreadable.
14) Results: Figure 7 – The description “CH” should be corrected to “HC”.
15) Results: The quality of Figures 8 - 12 should be improved.
16) The article's conclusion section is missing.
17) Abbreviations: Please check the abbreviations. It contains many abbreviations that do not appear in the body of the article, such as: CS, ET, UNECE, CVS, CoP, WHDC, NOxp, NOp.
18) References: Please check the correctness of the literature item [6]. Please also check that the literature formatting is correct with the journal requirements.
19) The manuscript contains language errors (e.g., in Table 1 – Denominations; decimal markers as commas). I suggest a proofreading of the manuscript.
Hope these comments can help you further improve the paper.
Comments on the Quality of English LanguageComments on linguistic correctness can be found in the body of the review.
Author Response
Dear Reviewer,
Please see our detailed response in attachment.
Best Regards,
Norbert Biro

Reviewer 2 Report
Comments and Suggestions for Authors
This paper is well written. It can be accepted after a minor revision.
My comments are as follows:
(1)Lack of the accuracy and measurement range of the emission measurement devices.
(2)What is the injection pressure in MPa?
(3)The text in Figures 6 to 12 is unclear.
Author Response

(The authors gave the same response as above.)

Reviewer 3 Report
Comments and Suggestions for Authors
The paper meets all the requirements of scientific work. The results of own measurements are presented in the paper. Appropriate statistical and graphic methods are used in their processing. I recommend publishing the article.
Author Response

(The authors gave the same response as above.)

Round 2
Reviewer 1 Report
Comments and Suggestions for Authors
Dear Authors,
Considering the corrections made to the manuscript, in my opinion, the manuscript in its current version can be accepted for publication in the journal Sensors.
Kind regards,
Author Response
Thank you very much for taking the time to review the article.
Best regards,
Norbert Biro